# Analysis of O-Ring Seal Failure under Static Conditions and Determination of End-of-Lifetime Criterion

**DOI:** 10.3390/polym11081251

**Published:** 2019-07-29

**Authors:** Anja Kömmling, Matthias Jaunich, Payam Pourmand, Dietmar Wolff, Mikael Hedenqvist

**Affiliations:** 1Bundesanstalt für Materialforschung und -prüfung, 12200 Berlin, Germany; 2KTH Royal Institute of Technology, Department of Fibre and Polymer Technology, School of Engineering Sciences in Chemistry, Biotechnology and Health, SE-100 44 Stockholm, Sweden

**Keywords:** EPDM, HNBR, seal failure, leak-tightness, DLO, oxygen permeability, DMA, indenter modulus

## Abstract

Determining a suitable and reliable end-of-lifetime criterion for O-ring seals is an important issue for long-term seal applications. Therefore, seal failure of ethylene propylene diene rubber (EPDM) and hydrogenated nitrile butadiene rubber (HNBR) O-rings aged in the compressed state at 125 °C and at 150 °C for up to 1.5 years was analyzed and investigated under static conditions, using both non-lubricated and lubricated seals. Changes of the material properties were analyzed with dynamic-mechanical analysis and permeability experiments. Indenter modulus measurements were used to investigate DLO effects. It became clear that O-rings can remain leak-tight under static conditions even when material properties have already degraded considerably, especially when adhesion effects are encountered. As a feasible and reliable end-of-lifetime criterion for O-ring seals under static conditions should include a safety margin for slight dimensional changes, a modified leakage test involving a small and rapid partial decompression of the seal was introduced that enabled determining a more realistic but still conservative end-of-lifetime criterion for an EPDM seal.

## 1. Introduction

Due to their excellent elasticity and resilience, seals made of elastomers are widely used in many technical applications to prevent leakage of fluids. However, like all polymers, elastomers age under the influence of e.g., oxygen, heat, time, UV radiation, and dynamic loads. For elastomer seals, both physical and chemical aging can deteriorate the seal performance. Physical aging involves reversible effects, e.g., relaxation through chain rearrangements [1,2], that lead to a loss of sealing force over time. On the other hand, chemical aging involves irreversible processes such as oxidation, chain scission, and additional crosslinking. The aging of elastomers such as hydrogenated nitrile butadiene rubber (HNBR) [3,4,5,6] and ethylene propylene diene rubber (EPDM) [7,8,9,10,11] as well as O-ring seals [12,13,14,15] has been investigated previously in numerous studies. During aging of EPDM, generally both chain scissions (mainly in the propylene segments [16]) and crosslinking reactions (mostly via the termonomer [8]) take place [17]. In HNBR, crosslinking reactions dominate [17,18]. Chain scission reactions lower the crosslink density, and thus result in softening and loss of elastic properties, while crosslinking during aging leads to stiffness increase and embrittlement of the material [19]. In order to determine the service lifetime of elastomer seals in accelerated aging experiments, an end-of-lifetime criterion has to be defined. This is especially relevant for applications in which the O-ring has to maintain its functionality over a large timespan (e.g., several decades) and regular replacements are impractical or impossible, e.g., in containers for radioactive waste. However, determining an end-of-lifetime criterion has proven to be difficult in the past. Given the many different applications of rubber, the standard ISO 11,346 [20] for estimating the lifetime of rubber cannot give clear guidance or values for an end-of-lifetime criterion, but mentions only that “the threshold value shall be chosen as the degree of degradation that is the maximum acceptable for the property being tested” and that 50% of the unaged value is commonly chosen. While many properties such as elongation at break or hardness can be measured on the aged material with a significant aging influence, the correlation with the sealing function remains unclear. Better indicators for the seal performance are compression stress relaxation measurements that monitor the sealing force decrease over time, and compression set measurements that relate to the remaining resilience. Using these methods, end-of-lifetime criteria of e.g., 80–85% compression set have been proposed in the literature [21,22,23,24]. However, for identifying the point of seal failure associated with a notable leakage rate increase, leakage rate measurements are necessary. Therefore, static leakage rate measurements were performed on HNBR and EPDM O-rings until significant leakage occurred after aging at 125 °C and 150 °C for up to 1.5 years. The failure mechanisms are analyzed and discussed in this paper.

However, O-rings can remain leak-tight under purely static conditions even when the sealing force approaches or equals zero [15,23,25] and the material properties have degraded considerably. For safe operation, such low sealing forces should not be reached, as the seal might be slightly disturbed by temperature changes (especially cooling, leading to thermal shrinkage), pressure changes, or vibrations. Therefore, a safety margin accounting for these cases should be considered when designating an end-of-lifetime criterion. This has been achieved in the present study with a modified leakage test involving a small fast partial release of the seal (from ~25% to ~23% compression in less than 1 s) that has been developed in our department [26]. As long as the seal is able to follow this partial decompression, sufficient resilience is left, and the seal can be operated safely. The gap size and decompression rate can be adjusted according to the specific application conditions and requirements. With this set-up, an end-of-lifetime criterion for an EPDM O-ring seal was determined and the correlation with other properties is discussed.

## 2. Materials and Methods

The studied commercial O-rings have an inner diameter of 190 mm and a cord diameter of 10 mm, which is relatively thick. The reason that we have chosen these samples is the background of our department, which is involved in the licensing procedures of casks for radioactive waste in Germany. In such casks, O-rings with a cord diameter of 10 mm are used. As the results of our research should be relevant for this specific application, we used O-rings with the same cord diameter. Additionally, these thick O-rings provided sufficient material for analysis after aging. Besides O-rings, 2 mm thick sheet material was aged, as this geometry is more suitable for many measurements of material properties. Investigated materials include hydrogenated nitrile butadiene rubber (HNBR) and ethylene-propylene-diene rubber (EPDM). HNBR is resistant against oils and used e.g., in automotive or oilfield applications, whereas EPDM exhibits a low glass transition temperature (around −50 °C) and is suitable for outdoor applications. The materials have an initial Shore A hardness of 80. Both materials are peroxide-cured and have an upper service temperature (UST) of 150 °C according to the manufacturer. The used EPDM contains a base polymer with 48 wt.% ethylene and 4.1 wt.% ethylidene norbornene (ENB), 90 phr of carbon black filler, and no plasticizer. The HNBR base polymer has an acrylonitrile content of 36 wt.% and an iodine number of 11 (corresponding to approximately 4% residual double bonds). The HNBR compound contains 80 phr of filler (mostly carbon black) as well as 5 phr of plasticizer. Due to confidentiality constraint, the detailed compound ingredients and producer cannot be provided here.

The samples were aged in air-circulating ovens at 150 °C and 125 °C and were examined after different exposure times (in days, d) of about 30 d, 100 d, 184 d (0.5 years), 365 d, and 545 d (1.5 years).

Dynamic-mechanical analysis (DMA) was performed in compression mode with a GABO (Ahlden, Germany) Eplexor 500 device on disks with 2.5 mm diameter that were die-cut from 2 mm thick aged sheets. Data obtained with a measurement frequency of 1 Hz are shown. The glass transition temperature was defined as the peak of the loss modulus E″ vs. temperature at 1 Hz measurement frequency. 

Oxygen permeability coefficients were determined using the time-lag method [27,28] with a constant volume/variable pressure set-up placed in a thermostated air bath (+/−0.1 K), equipped with a temperature-controlled pressure transducer (Baratron 621 C (MKS Instruments, Andover, MA, USA) with 10 mbar range. The measurement was performed on the topmost layer (thickness of 0.2–0.3 mm) that was cleaved from aged and unaged sheets of 2 mm thickness. Samples of 38 mm diameter were die-cut from these membranes and tested at 35 °C at 10 bar oxygen pressure after degassing in vacuum for four days.

Compressed half-O-rings were aged, which could be dismounted from the plates and analyzed after each aging time. Therefore, one sample was needed for each aging time and the corresponding number of samples was initially placed in the ovens for aging. On these samples, indenter modulus profiles were measured for detecting diffusion-limited oxidation (DLO) effects and characterizing the seal surface. The measurements were performed at 23 ± 1 °C and 50 ± 5% relative humidity using a modified Instron (Norwood, MA, USA) 5944 Universal Tensile Testing Machine, mounted with an Instron 10 N load cell, and a high-precision xy-stage providing travel increments of 0.05 mm in both the x and y directions. The indenter probe was constructed according to a previously used standard for a larger probe [12,29]. The 10 N load cell is very sensitive and can measure oscillating forces of up to 0.03 N without being in contact with the sample. Therefore, after reaching a contact force above 0.03 N and thus establishing contact with the sample, the load cell was balanced, and the force was set to zero automatically before each measurement. Then, the indenter tip penetrated the sample until a force of 0.05 N was measured. The force acting on the indenter probe was noted as a function of penetration depth. The indenter modulus (given in N/mm) was determined as the slope of the force–penetration–depth relationship between a force of 0.015 and 0.025 N. The data within this force range followed a strictly linear trend. Samples were scanned along both cross-section directions and across the mating surface with 0.5 mm increments. Three individual samples were tested for each aging state and all data points are displayed. The sample thickness varied between 3 and 7 mm. However, this did not influence the measurements, as the penetration depth was only about 0.02 mm and thus very small with respect to the sample thickness.

Three O-rings per material and temperature were aged with about 25% compression (corresponding to the actual compression in service) between flanges in fixtures for leakage rate measurements (Figure 1). After a specific aging time, the O-rings were tested according to the procedure described below, and then placed back in the ovens for further aging. As the bolts of the fixtures remained tightened, the compression of 25% remained unchanged throughout the whole aging time and leakage rate measurements. The seals were dismounted only after seal failure had occurred. Table 1 illustrates the sample nomenclature. For most seals, no lubricant was used, as it was not necessary for correct assembly. 

The leakage rate *Q* is a quantity describing the change in pressure ∆*p* within a specific volume *V* in a specific time period ∆*t* and is thus calculated according to Equation (1).
(1)$$Q=\frac{\Delta p\cdot V}{\Delta t}$$

Leakage rate measurements were performed using the pressure-rise method with a set-up shown schematically in Figure 2. The volume inside the O-ring between the flanges (plus the volume of the hose) is designated as R1. First, valve V2 was closed and the remaining volume was evacuated to 10^−2^ mbar for 8 h by the pump P for degassing the O-ring. Afterwards, valve V1 was closed and V2 opened to release the air of the test volume R2 into the total volume (R1, R2, and pipes/hoses). From the resulting pressure (measured by sensor S1 with a working range from 10^−2^ to 110 mbar) the relevant volume *V* for leakage rate calculation was determined with the ideal gas law. Next, V1 was opened and the total volume *V* was evacuated down to 10^−2^ mbar. Then, V1 was closed and the pressure rise in the total volume *V* was measured over two hours with sensor S2, which has a working range from 10^−3^ to 11 mbar. The leakage rate was calculated from the pressure difference measured in these two hours. As metallic seals were used to seal the pipe transitions of the set-up, the measured leakage rate is assumed to be only due to leakage at the O-ring. All static measurements were performed at 20 °C, 60 °C, and −30 °C (HNBR) or −40 °C (EPDM). 

Besides these static pressure rise tests, the pressure rise during and after rapid partial release was tested at 20 °C using a device developed in our department that is described in detail in [26]. The device was designed for testing the low-temperature leak-tightness but can also be applied for aged samples. It enables a release of the studied O-ring by approximately 0.2 mm from 25% to 23% compression in less than one second during the pressure rise measurement. If the resilience of the seal has decreased below a certain value, the seal cannot follow the decompression fast enough and a leakage path can open up. EPDM seals were aged in a special fixture designed for this measurement and were tested by first measuring the static pressure rise, then evacuating the inside of the fixture again, and measuring the pressure rise during and after partial release. One data point per second was recorded during the static test, and one data point every 0.4 s during the dynamic test. After evacuation, data recording is activated. Then, the valve to the pump is closed, starting the pressure rise measurement. Subsequently, the partial release is applied with a wrench and the pressure is recorded for up to 45 min. If the seal remained leak-tight, it was recompressed to 25% and continued aging.

The performed experiments are summarized in Table 2.

## 3. Results and Discussion

### 3.1. Dynamic-Mechanical Analysis (DMA)

Figure 3 shows storage modulus E′ and loss modulus E″ of the HNBR sheet material (chosen because O-rings exhibited DLO effects, see next section). The storage modulus in the rubber plateau region increased with aging time. Furthermore, the temperature region where a drop in the storage modulus and a peak in the loss modulus is observed increased with increasing aging time. This indicated an increased glass transition temperature and hardening due to the dominant crosslinking reactions occurring during aging of HNBR [17,18]. Beyond 98 d at 150 °C and 367 d at 125 °C, the material was too brittle to prepare samples for the measurement. At −30 °C, which is the lower temperature for leakage rate measurements in Section 3.3, the material is in the glassy state.

Figure 4 shows storage modulus and loss modulus of EPDM sheet material. Similar to HNBR, the rubber plateau of the storage modulus increased, and the glass transition shifted to higher temperatures with increased aging exposure. This indicates that crosslinking is the main degradation mechanism in EPDM as well, but the changes are not as pronounced as for HNBR. After 185 d at 150 °C, the intensity of the glass transition is dramatically reduced, as the chain mobility has decreased considerably due to the increased crosslink density, and the material acts more like a densely crosslinked thermoset than a loosely crosslinked rubber. At −40 °C, which is the lower temperature for leakage rate measurements (cf. Section 3.3), the unaged EPDM material is in the glass transition region between glassy and rubbery state.

### 3.2. Permeability

Figure 5 shows the oxygen permeability of EPDM and HNBR. The permeability of unaged HNBR is about 5 times lower than that of EPDM (probably due to the higher polarity of HNBR [30]) and has decreased notably after 10 d of aging at 150 °C [25], owing to the dominant crosslinking reactions during aging [17,18]. For EPDM, the permeability decreased significantly after aging for 34 d at 150 °C. Similar to the DMA results, EPDM shows the same trend as HNBR (reduced permeability due to crosslinking), but the changes occur slower during aging. 

### 3.3. Static Leakage Rate Measurements

#### 3.3.1. HNBR

Figure 6 shows results from static leakage rate measurements of HNBR O-rings aged at 150 °C and 125 °C. Two additive effects contribute to the measured leakage rate: Gas permeation through the material, and actual leakage between O-ring and flange. As long as the O-ring is intact and functioning, mainly permeation occurs. As the gas permeability of polymers increases with temperature, the measured leakage rate is usually higher for higher temperatures, as is the case in Figure 6. Furthermore, a decrease of the measured leakage rate after aging is observed, especially for 30 and 100 days of aging time. The probable reason for this are the dominant crosslinking reactions during aging of HNBR [17,18], which lead to an increase of crosslink density and thus a decrease of permeability. Even though the sealing force has decreased strongly [25] and material properties have altered dramatically, as is visible in the DMA measurements in Section 3.1, the leakage rate has hardly changed after 98 d at 150 °C and 365 d at 125 °C, However, DLO effects in the thicker O-rings prevent homogeneous and overall degradation, in contrast to the 2 mm thin sheet material used for the DMA measurements. Therefore, the strong material property degradation observed in the DMA measurements occurs only at the outer region at the sides of the O-ring, while a large volume of the O-ring has not degraded as strongly and can maintain the sealability (cf. Figure 11). After 184 days (half a year) at 150 °C, the first leakage (cf. number 1 in Figure 6a) occurred at O-ring N1 at −30 °C and is marked by the steeply ascending dashed line (representing a possible development between 100 and 184 days) near the number 1 in Figure 6a. Before failing at −30 °C, the seal N1 exhibited an increased leakage rate at 20 °C (cf. number 2 in Figure 6a), probably indicating a change of the contact conditions that led to failure at the lower temperature. Furthermore, the other two seals (N2 and N3) exhibited an increased leakage rate at −30 °C after 184 days and failed after 224 days of aging time (cf. number 3 in Figure 6a). However, after failing for the first time at the lower temperature, the seals became leak-tight again when raising the temperature to 20 °C. This means that the increased leakage rate was only due to reversible thermal shrinkage. 

The leakage rates of seals N4, N5, and N6 that were aged at 125 °C are shown in Figure 6b. All seals remained leak-tight at 60 °C, 20 °C, and −30 °C for up to 365 days/1 year of aging with only slight changes of the measured leakage rates compared to the unaged values. At the next measurement after 545 d (1.5 years), all three seals were completely non-tight, i.e., the measurement volume could not be evacuated sufficiently. In order to investigate the behavior at low temperatures in more detail, an unaged seal and N6 were measured at additional low temperatures of −26 °C, −28 °C, −32 °C, −34 °C, and −36 °C as shown in Figure 7. Both one of the unaged seals and N6 aged for 365 d exhibited an increased leakage rate at −36 °C. For N6, the leakage rate at −36 °C after aging for 365 d is also shown by the single blue triangle in Figure 6b. However, N6 aged for 98 d and 184 d remained leak-tight down to −36 °C, perhaps due to a better adaption and sticking of the seal to the flanges after being exposed to the influence of time and elevated temperature. When disassembling O-rings after having been aged in the compressed state between flanges, aged seals often stuck to the flanges and were sometimes hard to detach. Therefore, the sticking effect probably has a significant influence on the sealing behavior. However, this sticking effect only occurred when no lubricant was used, which acts as a separator between the rubber and the flange plates. The influence of using a lubricant is discussed in Section 3.5. The good adhesion between seal and flange is also obvious in Figure 12a, where the work grooves of the metal flange are visible on the O-ring in circumferential direction.

It is expected that seal failure occurs first at the lowest measuring temperature, as the real compression of the O-ring and, thus, the sealing force is lower due to thermal shrinkage. Additionally, at the lowest measurement temperature (−30 °C), the material is in the glassy state (cf. Section 3.1) and no longer supple, which means that the thermal shrinkage cannot be compensated by elastic recovery. For example, when the O-ring dimension is 10 mm at 23 °C and the space between the flange plates is 7.5 mm, 25% compression of the O-ring results. With a thermal expansion coefficient of 1.25 × 10^−4^/K, the O-ring shrinks by about 0.66% at −30 °C, with a resulting compression of 24.5%. Moreover, the HNBR seals shrink during aging due to the dominant crosslinking reactions. On O-rings that had aged in the uncompressed state, the shrinkage was about 2% after 184 d at 150 °C and about 1.5% after 365 d at 125 °C. However, especially the value at 150 °C is affected by DLO effects, as the outer, more strongly aged region exhibits a higher shrinkage than the less-aged interior. If the O-ring had aged homogeneously, the measured overall shrinkage would have been higher.

#### 3.3.2. EPDM

Figure 8a shows results from static leakage rate measurements of EPDM O-rings aged at 150 °C. For the measured aging times of 30 and 100 d, the leakage rate hardly changed at 20 °C and 60 °C compared to the unaged values, and decreased slightly at −40 °C, possibly due to sticking and, thus, better contact between seal and flange. At the next aging step after 184 d (0.5 years), all three seals were non-tight at all temperatures. The longer lifetime of HNBR seals aged at 150 °C in comparison to EPDM seals aged at 150 °C is due to the pronounced DLO effects of HNBR O-rings aged at 150 °C that limit aging mainly to the outer regions (cf. Section 3.4), thus preserving the sealability. When comparing HNBR and EPDM seals aged at 125 °C, where DLO effects are less pronounced, EPDM seals exhibited a longer lifetime, as all three EPDM seals remained leak-tight up to 545 d aging time when measured at 20 °C and 60 °C (Figure 8b). However, after 545 d (1.5 years), two seals (E5 and E6) became non-tight at −40 °C (number 1 in Figure 8b, the dashed line represents a possible course between 365 d and 545 d). Seal E4 became non-tight when heating again to 20 °C after measuring at −40 °C (number 2 in Figure 8b). After 365 d, E4 exhibited atypical behavior with measured leakage rates deviating from the other two seals. It is not clear what caused this and if it influenced the leak-tightness of E4 after 545 d in contrast to the other two seals. However, after 365 d, the material is already strongly degraded with a compression set near 100% [25] (i.e., almost no ability to recover), so perhaps additional unforeseen effects occurred.

Similar to HNBR, EPDM seals remained leak-tight even though material properties (e.g., measured with DMA) and sealing force [25] had already changed notably after 98 d at 150 °C. However, seals aged for 545 d at 125 °C became non-tight at low temperatures, even though aging for 550 d at 125 °C led to less changes of the DMA properties than aging for 98 d at 150 °C. The reason for this discrepancy is probably DLO effects which reduce aging at the seal face of the O-ring aged at 150 °C (cf. Figure 13), thus leading to apparently longer lifetimes.

In order to investigate in more detail the behavior at low temperatures, where seal failure is most likely, selected EPDM seals were also measured at additional low temperatures of −50 °C, −54 °C, −56 °C, −58 °C, and −60 °C, see Figure 9. During cooling from −50 °C to −60 °C, EPDM became fully glassy (cf. Figure 4). The leakage rate decreases with decreasing temperature and drops especially fast during the glass transition below −40 °C due to the significantly reduced permeability of the glassy material [31]. At the aging steps that were measured, the seals were leak-tight at all measured temperatures down to −60 °C.

The leak-tightness at low temperatures was also investigated on different (unaged) materials by us previously [32].

### 3.4. Influence of Diffusion-Limited Oxidation (DLO) Effects on Leak-Tightness

When doing accelerated aging experiments, diffusion-limited oxidation (DLO) effects can occur depending on aging temperature and time, sample thickness, and oxygen permeability of the material. DLO effects appear when the outside of the sample, where oxygen availability is good, continues to age, while less oxygen diffuses to the interior, with the result that oxidative aging is slowed down there. This leads to heterogeneously aged samples (cf. Figure 10a) that yield distorted bulk properties, e.g., compression set, but can also affect leak-tightness, as will be discussed below.

In previous papers [25,33], we already investigated DLO effects in our samples in detail. For HNBR aged at 150 °C, DLO effects are noticeable already after 10 d, and from 100 d onward, there is a very strong oxidation gradient between interior and outer parts of the sample. At 125 °C, DLO effects are also measurable, but less pronounced and thus with limited impact on the measured bulk properties. For EPDM, DLO effects could be detected after 100 d of aging at 150 °C and became pronounced after 185 d. At 125 °C, no DLO effects were observed for EPDM for up to 2 years (730 d) aging time. The more pronounced DLO effects of HNBR in comparison to EPDM are probably due to the lower oxygen permeability of unaged HNBR (cf. Figure 5), the faster oxidation of HNBR due to unsaturated double bonds in the backbone, and the faster decrease of the permeability during aging due to dominant crosslinking in HNBR (cf. Figure 5).

In order to correctly interpret the leakage rate measurements and seal failure in the previous section, modulus indenter measurements were performed on the O-ring cross-section and the sealing surface of O-rings (Figure 10b) which were aged in compression and disassembled for analysis (in contrast to the O-rings in fixtures for leakage rate measurements, which continued aging after each leakage rate measurement and were only disassembled after seal failure). The measured modulus was regarded as an indicator for oxidative degradation. The measurements were performed on strongly aged samples before or at the aging state at which seal failure occurred in the leakage rate measurements (101 and 185 d at 150 °C, respectively). The results are shown in Figure 11 for HNBR and in Figure 13 for EPDM.

After 101 d aging time at 150 °C, the indenter modulus profiles show pronounced DLO effects in x-direction with much higher moduli near the surface. In y-direction, however, the modulus is uniformly low, as oxygen access was limited both by the long diffusion path from the sides and by the flanges on the top and bottom. Across the seal face, the modulus is slightly higher near the sides, where oxygen access is better. However, as the oxidatively aged regions of HNBR undergo additional crosslinking and thus hardening and shrinkage (cf. end of Section 3.3.1), a gap between the O-ring and flange opens from each side as aging proceeds. Therefore, oxidation can advance from both sides until only a very narrow region remains with low modulus after 185 d. This region with low modulus still retains rubbery properties, which maintain the sealing function. However, this small contact area between the rubbery O-ring and flange is very delicate and can only maintain leak-tightness under static conditions. Even slight dimensional changes induced by thermal shrinkage can lead to contact loss, as the recovery of the narrow rubbery region is constrained by the hardened surrounding material. Thus, when cooled to −30 °C, the first O-ring failed after 185 d aging time at 150 °C, as shown in Figure 6, and the other two O-rings failed shortly afterwards (224 d) at −30 °C. The remaining ability of the narrow rubbery region to recover after decompression can even be seen in a photo of seal N2 (Figure 12a) after it had been disassembled in consequence of its failure at −30 °C after 224 d aging time at 150 °C. Furthermore, the O-ring surface on the sides was covered by fine cracks (probably due to shrinkage and embrittlement) and several large cracks could be seen as well. The large cracks reaching onto the seal face were probably the locations where leakage occurred. By contrast, strongly aged (1 year at 150 °C) EPDM O-rings exhibited one deep crack at the outer circumference (Figure 12b) when aged in the uncompressed state.

For EPDM, only slight DLO-effects were measurable in x-direction after aging for 101 d at 150 °C (Figure 13). Similar to HNBR, the modulus is uniform in y-direction as oxygen access is hindered from all sides. As explained for HNBR, additional crosslinking during aging leads to shrinkage and thus oxidation and hardening can proceed from both sides of the seal face. After 101 d at 150 °C, only a narrow region with low modulus remains on the seal face. At this aging state, the O-rings were still leak-tight, even when cooled to −60 °C (cf. Figure 9). However, when measured after aging for 185 d at 150 °C, all three O-rings were completely non-tight (cf. Figure 8a). This is also obvious from Figure 13b: As seal and flange are no longer in contact, oxidation can also proceed from the seal face, leading to an increased modulus at the top and bottom in y-direction.

### 3.5. Influence of Sticking/Lubricant

As shown in the previous sections, O-rings could remain leak-tight even if only a narrow rubbery region on the seal face remained. For EPDM, this was even the case when the temperature was lowered to −60 °C, which is a large temperature difference of 80 °C to room temperature, resulting in thermal shrinkage by 1.6% (*α* = 2 × 10^−4^/K). An effect that was probably advantageous for leak-tightness is sticking. When disassembling half O-rings that had aged in the compressed state after specific aging times (30 d, 100 d, 185 d, etc.), the O-rings usually stuck to the flanges and were often hard to detach. EPDM stuck especially strongly, so that often considerably high force had to be used and some parts even ripped off the material and clung to the flange, see Figure 14. The reason for this is the relatively large degree of chain scissions occurring during aging of EPDM [16] that make the material sticky [34], e.g., through the formation of oxygen containing polar groups.

In order to investigate the influence of sticking, additional experiments were conducted using lubricant (silicon oil) as separating layer between seal and flange. A lubricant is used commonly for O-ring assembly e.g., in dovetail grooves or other more complex assembly geometries. Three EPDM seals were aged at 150 °C to obtain significant results with the shortest aging times. Every seal was measured down to −60 °C at each aging step. The temperature sequence was varied on each O-ring in order to detect a possible influence and is shown in Table 3. After measuring first at 20 °C, seals E7 and E9 were measured at 60 °C and then at lower temperatures, while E8 was measured first at the lower temperatures. The sequence of E9 is the same as the one used for the low temperature tests of the non-lubricated seals. When a seal exhibited an increased leakage rate at lower temperatures, it was measured again at 20 °C to see whether the leak was only due to the reversible effects of variation in temperature or a permanent feature. If the seal became tight again at 20 °C, aging was continued.

When comparing lubricated and non-lubricated seals after approximately 30 days of aging at 150 °C, the lubricated seals exhibited lower leakage rates, but higher variability, as shown in Figure 15. According to [35], lubricants help to reduce the leakage rate in vacuum applications (as in our pressure rise experiment) by filling the microfine inclusions of the flange’s metal surfaces and lowering permeation rates of the elastomer. The higher variability between the measurements is probably due to the different extents of wetting of the O-ring surface, as only the flange was sprayed with the lubricant oil, and not the whole O-ring.

The leakage rate measurements performed on lubricated seals after 31 d, 67 d, 94 d, and 125 d aging time at 150 °C are shown in Figure 16. Similar to Figure 9, the leakage rate decreases with decreasing temperature and drops faster below −50 °C in the vicinity of the glass transition region. As the glass transition region broadens and the glass transition temperature increases from about −55 °C to −43 °C after aging for approximately 100 d at 150 °C (cf. Figure 4), the faster decrease of the leakage rate is already observed between −40 °C and −50 °C for the sample E7.

After 94 d aging time at 150 °C, the seal E7 first became non-tight at −58 °C and seal E8 exhibited a higher leakage rate when lowering the temperature to 0 °C and below. E9 showed strongly increased leakage rate at −58.6 °C already after 67 d of aging at 150 °C. Thus, the lubricated seals became non-tight much earlier compared to the non-lubricated seals, which all remained leak-tight down to −40 °C (and in one measured case down to −60 °C) after aging for 98 d at 150 °C. This reveals the significant influence of sticking on the leak-tightness.

On the other hand, an influence of the temperature sequence for leakage rate measurements was not observed. Seals E7 and E9 were both first measured at the higher temperature (60 °C), but E7 stayed leak-tight the longest, and E9 the shortest.

### 3.6. Determination of End-of-Lifetime Criterion

As outlined in the introduction, the standard ISO 11,346 [20] for estimating the lifetime of rubber includes no clear guidelines or values for identifying a suitable end-of-lifetime criterion for O-ring seals, and most values for end-of-lifetime criteria from the literature are based on the degradation of material properties (e.g., elongation at break) or sealing properties (e.g., compression set). However, as leakage rate is the measure directly related to seal performance, an end-of-lifetime criterion for O-ring seals should be based on the occurrence of significant leakage. Therefore, the original idea was to determine an end-of-lifetime criterion based on a significant increase of (static) leakage rate indicating the end of serviceability of the seal. However, as we have seen, elastomer O-rings can remain leak-tight under static conditions even when material properties have already degraded considerably [15,23,25] and when only a narrow strip of rubbery material remains on the seal face. Therefore, when taking into account a safety margin for vibrations, pressure changes or thermal shrinkage, it becomes obvious that a feasible and reliable end-of-lifetime criterion has to be chosen well before the point of seal failure observed in the static leakage measurements.

In order to determine an end-of-lifetime criterion for static seals with a sufficient safety margin, a modified leakage test involving a small and rapid partial decompression of the seal has been developed at our department (originally for measurements at low temperatures [26]) and adapted for measurements on aged seals. It includes a release of the compressed seal by about 0.1 to 0.2 mm (corresponding to 1% to 2% compression for the investigated seal dimension) in less than one second. Thus, the requirements for the seal are much higher compared to the static case, as the seal has to follow the decompression fast enough with elastic recovery without losing contact. The time and distance for the decompression can be varied based e.g., on boundary conditions or accident scenarios for specific applications.

The experiments were performed on non-lubricated EPDM seals. Based on the findings in Section 3.5, it would have been a more conservative approach to use lubricated seals, but that was not known at the start of the experiments. Generally, the experiment presented in the following does not have the aim of identifying a universal value for an end-of-lifetime criterion, but it suggests a method that can be used to obtain a criterion for specific application conditions. 

The EPDM seal was aged with 25% compression in the adapted flange system (described in detail in [26]), then the pressure rise after evacuation was measured at 20 °C with the standard (static) procedure, and after anew evacuation with the partial decompression. Figure 17 shows the results of the first test. For 32 d, 70 d, and 101 d aging time at 150 °C, the O-ring remained leak-tight for the static case, as was expected from the previous leakage rate measurements (cf. Figure 8a). When the partial release was applied during the measurement, the O-ring remained leak-tight only for up to 70 d aging time (Figure 8b). When tested at the next aging step after 101 d, a strong pressure increases up to ambient pressure was observed when the partial release was applied, thus indicating seal failure.

As the time between 70 d and 101 d is relatively long, the experiment was repeated with a seal aged for 56 d, 70 d, and 80 d at 150 °C. As expected, the seal remained leak-tight also in the dynamic test for up to 70 d aging time, but became non-tight after 80 d [36], and thus much earlier than in the static case. As it is not clear at which aging time between 70 d and 80 d leakage would first occur, the aging state after 70 d at 150 °C is conservatively assumed as the end of the lifetime. At roughly this aging condition (67 d), the lubricated seals were also all still leak-tight in the static case when measured at 20 °C. When performing this experiment, it is important to avoid DLO effects, which lead to different results than samples that age homogeneously (as would be the case at lower service temperatures). For the investigated EPDM seal aged at 150 °C, DLO effects should be small after 70 d at 150 °C, as no DLO effects were observed after 30 d, and only slight effects after 100 d (cf. Figure 13 and [25,33]). A new experiment is currently being performed on an EPDM seal aging at 125 °C to exclude DLO effects completely. The seal remained leak-tight in both test set-ups at the latest test after 248 d aging time, which corresponds to approximately 69 d at 150 °C when employing the time–temperature superposition shift factor determined with compression set data in [25].

The aging state at which the end of the lifetime was defined (70 d at 150 °C for this EPDM seal) can be correlated to other methods as shown in Table 4. Except for the continuous compression stress relaxation measurement, the values shown in Table 4 were interpolated between data measured after 30 and 100 d aging time. The strongest indication of degradation is observed in compression set, compression stress relaxation, and elongation at break with about 85% to 90% change compared to the values for the unaged samples. On the other hand, the maximum loss factor measured by DMA changed by 46%, glass transition temperature by 5 °C, and hardness by 6 units. The results illustrate that the choice of the method that is used to investigate aging effects makes a huge difference. For seals, properties measured in the compressed state (set, relaxation) are especially relevant and they also exhibit the strongest degradation. Furthermore, it should be mentioned that the measurement of elongation at break on standard tensile samples yielded a similar result. 

However, these findings are only valid for the tested material and the observed correlations have to be verified with other materials. Additionally, the end of the lifetime determined by using the modified leakage test is probably specific for the tested geometry. 

## 4. Conclusions

The leakage rate of elastomer O-rings hardly changes during aging before failure occurs. The seals can remain leak-tight under static conditions even at low temperatures (when they are in the glassy state) and despite considerable deterioration of the material properties. When DLO effects are involved, aging of the mating surfaces proceeds from the sides until the area of the seal face is completely hardened. An effect that contributes to the apparently long seal lifetime before leakage is sticking of the O-ring to the flanges. Therefore, when a lubricant was applied as a separating layer between rubber and metal, leakage occurred much earlier. In order to determine an end-of-lifetime criterion for O-ring seals that is correlated to leakage rate as direct indicator of seal performance, but more conservative than the point of static seal failure, a modified leakage test involving a small and rapid partial decompression of the seal during the leakage rate measurement was introduced. This test led to an earlier failure of the investigated EPDM seal. When correlating the lifetime obtained from this test to other properties, compression set, compression stress relaxation, and elongation at break exhibited similar notable changes by about 85% to 90% for this aging state. The method presented here can be adapted to different materials, geometries, and application conditions for determining a specific end-of-lifetime criterion for O-ring seals. 

## Figures and Tables

**Figure 1 polymers-11-01251-f001:**
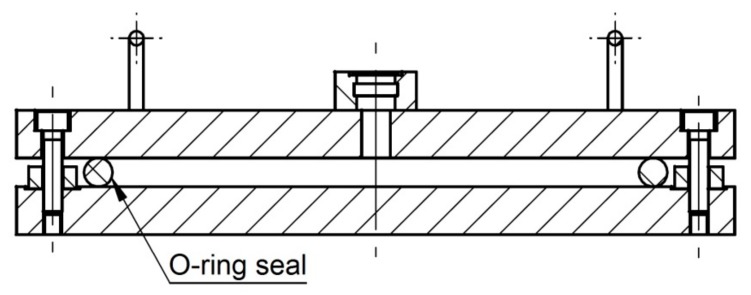
O-ring in fixture for leakage rate measurements before compression by 25%.

**Figure 2 polymers-11-01251-f002:**
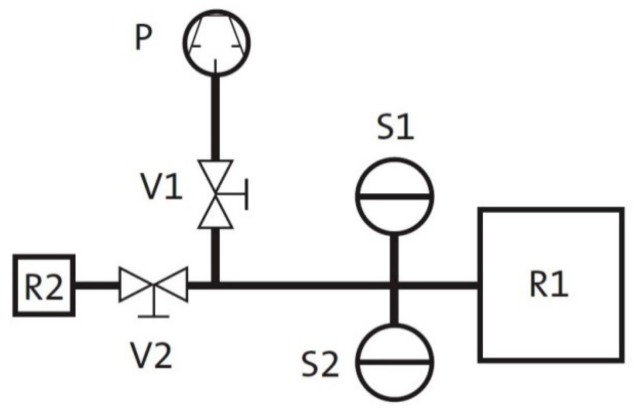
Set-up for leakage rate measurements.

**Figure 3 polymers-11-01251-f003:**
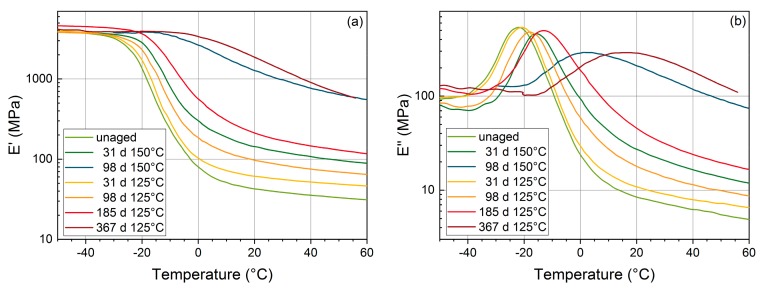
Storage modulus E′ (**a**) and loss modulus E″ (**b**) of hydrogenated nitrile butadiene rubber (HNBR) vs. temperature measured on sheet material at 1 Hz measurement frequency.

**Figure 4 polymers-11-01251-f004:**
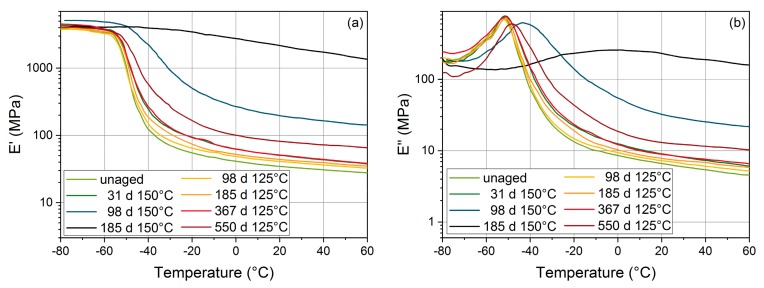
Storage modulus E′ (**a**) and loss modulus E″ (**b**) of ethylene propylene diene rubber (EPDM) vs. temperature measured on sheet material at 1 Hz measurement frequency. Note that the graphs for 31 d at 150 °C and 367 d at 125 °C are almost identical.

**Figure 5 polymers-11-01251-f005:**
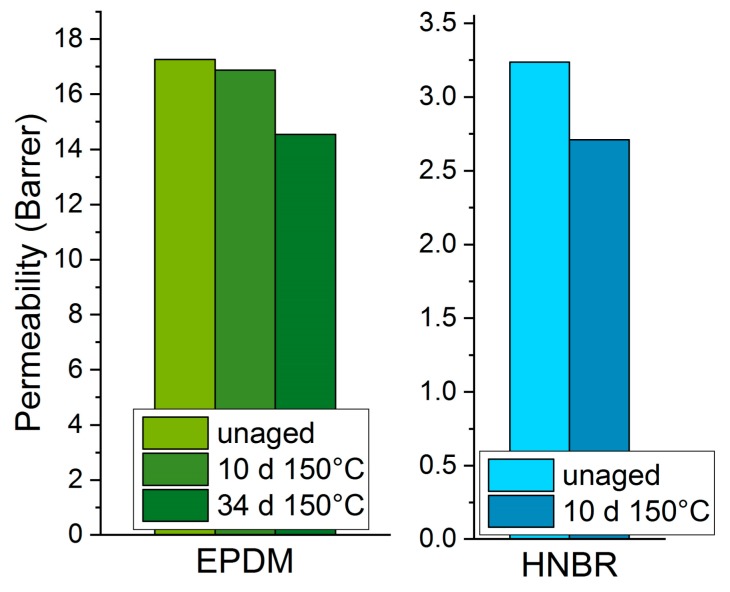
Oxygen permeability of EPDM and HNBR. Note the different scales for each material.

**Figure 6 polymers-11-01251-f006:**
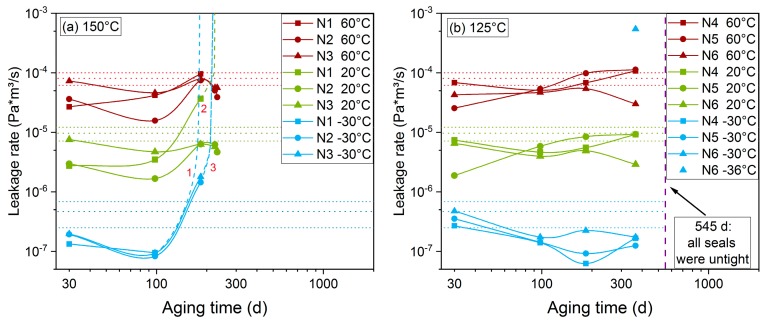
Static leakage rate measured at 60 °C, 20 °C, and −30 °C on HNBR O-rings aged at (**a**) 150 °C and (**b**) 125 °C. The dotted horizontal lines represent the average value of three unaged O-rings for each test temperature and the respective standard deviation range.

**Figure 7 polymers-11-01251-f007:**
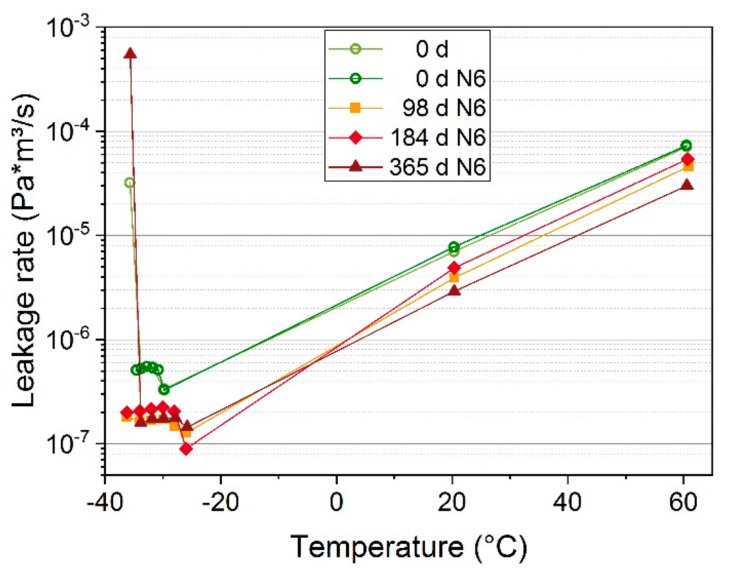
Leakage rate of an unaged seal and the seal N6 aged at 125 °C.

**Figure 8 polymers-11-01251-f008:**
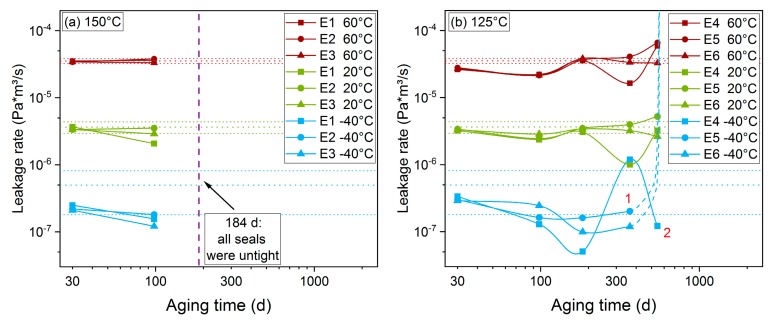
Static leakage rate measured at 60 °C, 20 °C, and −40 °C on EPDM O-rings aged at (**a**) 150 °C and (**b**) 125 °C. The dotted horizontal lines represent the average value of three unaged O-rings for each test temperature and the respective standard deviation range.

**Figure 9 polymers-11-01251-f009:**
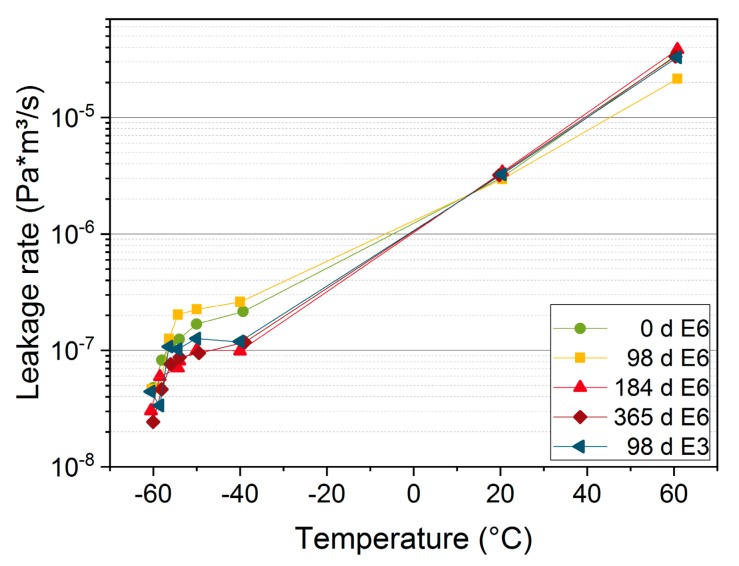
Leakage rate of the seal E6 aged at 125 °C and E3 aged at 150 °C for the respective aging periods vs. temperature.

**Figure 10 polymers-11-01251-f010:**
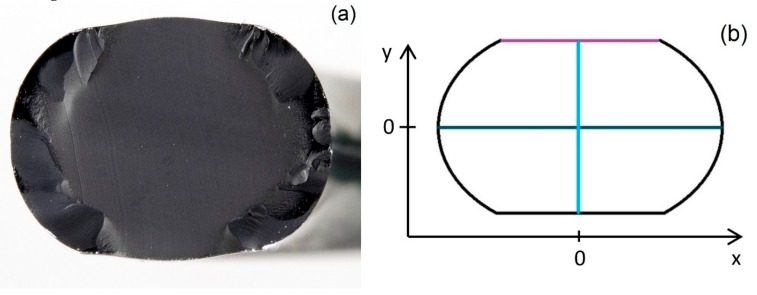
(**a**) Photo of the cut cross-section of an HNBR O-ring with compression set and DLO effects aged for 101 days at 150 °C. (**b**) Drawing of the cross-section of an O-ring with compression set. The colored lines represent the respective measurement lines for the indenter modulus measurements shown in Figure 11 and Figure 13.

**Figure 11 polymers-11-01251-f011:**
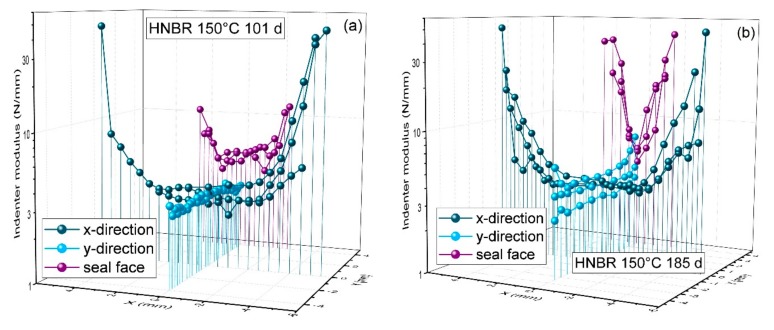
Indenter modulus profiles on the O-ring cross-section in x-direction (dark blue), in y-direction (light blue), and across the seal face in x-direction (purple) determined on HNBR O-rings aged at 150 °C for (**a**) 101 days and (**b**) 185 days (half a year).

**Figure 12 polymers-11-01251-f012:**
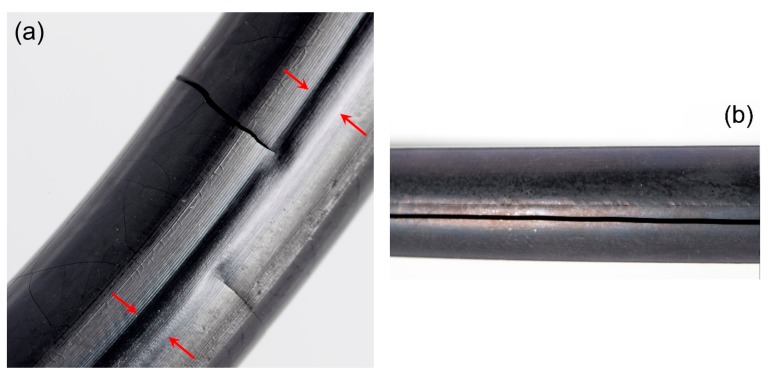
(**a**) Photo of seal N2 after failure and aging for 224 days at 150 °C. Red arrows indicate the recovered narrow rubbery region on the seal face. (**b**) Photo of EPDM O-ring aged in the uncompressed state for 1 year at 150 °C with a deep circumferential crack.

**Figure 13 polymers-11-01251-f013:**
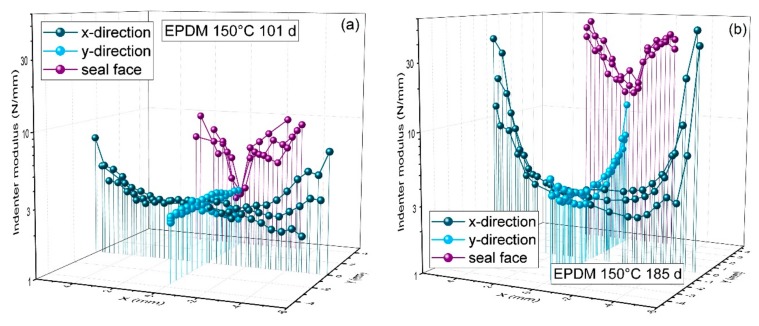
Indenter modulus profiles on the O-ring cross-section in x-direction (dark blue), in y-direction (light blue), and across the seal face in x-direction (purple) determined on EPDM O-rings aged at 150 °C for (**a**) 101 days and (**b**) 185 days (half a year).

**Figure 14 polymers-11-01251-f014:**
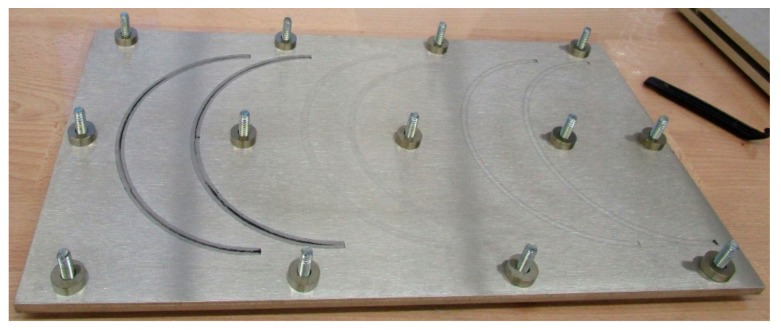
Flange on which two half O-rings made of EPDM (left), FKM (fluorocarbon rubber, middle) and HNBR (right) had aged in the compressed state for 182 d at 125 °C.

**Figure 15 polymers-11-01251-f015:**
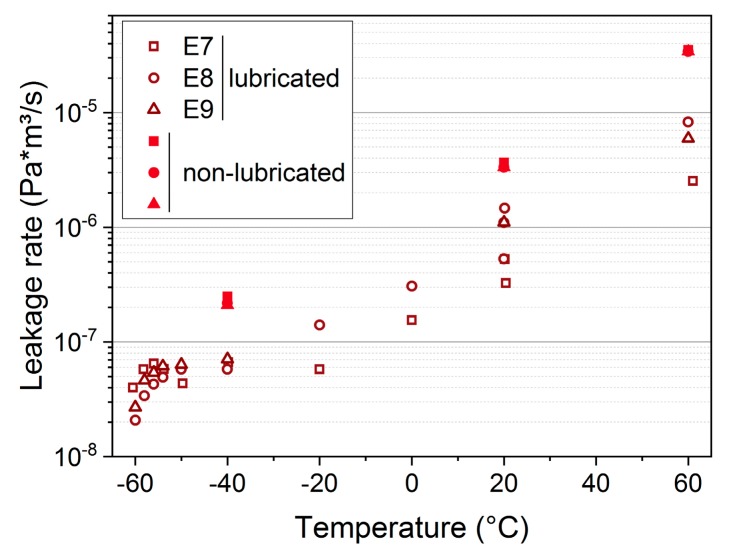
Leakage rates of lubricated (with silicon oil) and non-lubricated EPDM seals aged for 30 d at 150 °C.

**Figure 16 polymers-11-01251-f016:**
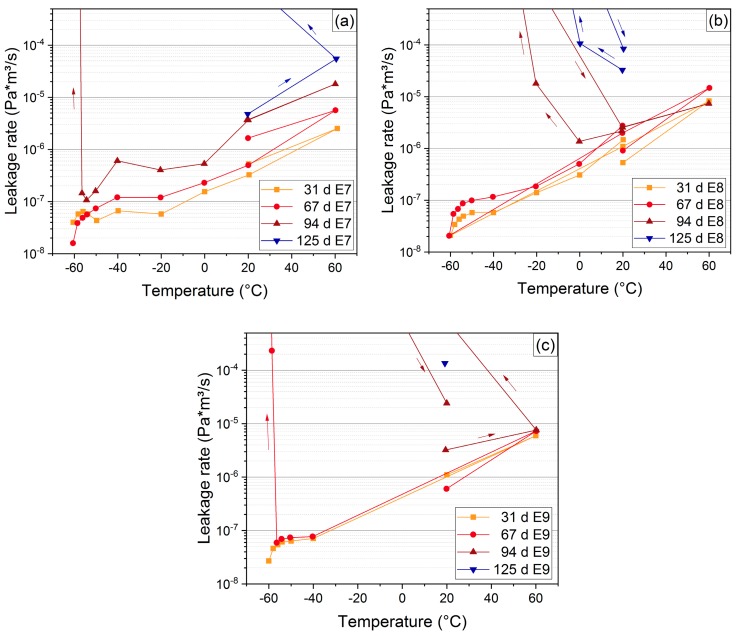
Measured leakage rates vs. temperature for lubricated EPDM seals E7 (**a**), E8 (**b**), and E9 (**c**) at 150 °C. The arrows indicate the temperature sequence (cf. Table 3).

**Figure 17 polymers-11-01251-f017:**
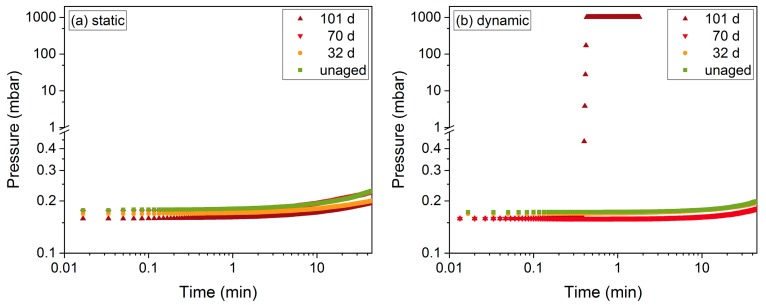
Pressure rise during (**a**) static and (**b**) dynamic leakage test (with partial decompression) of an EPDM seal aged at 150 °C for the respective aging times.

**Table 1 polymers-11-01251-t001:** Overview of seal nomenclature.

Seal Label	Material	Aging Temp.	Lubricated
N1, N2, N3	HNBR	150 °C	No
N4, N5, N6	HNBR	125 °C	No
E1, E2, E3	EPDM	150 °C	No
E4, E5, E6	EPDM	125 °C	No
E7, E8, E9	EPDM	150 °C	Yes

**Table 2 polymers-11-01251-t002:** Overview of experiments and samples.

Measurement	Material	Aging Temp.	Aging Times (+Unaged)	Sample Geometry
DMA	HNBR	125 °C, 150 °C	30 d to 1 a	2.5 mm diameter disks from 2 mm thick sheets
	EPDM	125 °C, 150 °C	30 d to 1.5 a	
Oxygen permeability	HNBR	150 °C	10 d	0.2–0.3 mm thick sheets with 38 mm diameter
	EPDM	150 °C	10 d, 30 d	
Indenter Modulus	HNBR, EPDM	150 °C	100 d, 0.5 a	half O-rings aged in compression
Static leakage rate	HNBR, EPDM	125 °C, 150 °C	30 d to 1.5 a	O-rings aged in compression
Static leakage rate with lubricated seals	EPDM	150 °C	30 d to 125 d	
Leakage rate with partial decompression	EPDM	150 °C	30 d to 100 d	

**Table 3 polymers-11-01251-t003:** Temperature (°C) sequence for leakage rate measurements performed on lubricated EPDM seals E7, E8, and E9 aged at 150 °C.

E7	E8	E9
20	20	20
60	0	60
20	−20	−40
0	−40	−50
−20	−50	−54
−40	−54	−56
−50	−56	−58
−54	−58	−60
−56	−60	
−58	20	
−60	60	

**Table 4 polymers-11-01251-t004:** Correlation of the aging state defined as the end of the lifetime with different methods [37].

Change after 70 d at 150 °C	Method/Property	Measurement/Standard	Sample Geometry
+89%	Compression set	DIN ISO 815-1 [38]	Half O-rings
−86%	Compression stress relaxation	DIN ISO 3384 [39]	4 cm O-ring segments
−91%	Elongation at break	DIN EN ISO 527-1 ^1^ [40]	S2 samples
−46%	Maximum loss factor	DMA peak tan *δ* at 1 Hz	2.5 mm diameter disks from 2 mm thick sheets
+5 °C	Glass transition temperature	DMA peak E″ at 1 Hz	
+6	Shore A hardness	DIN EN ISO 868 [41]	Uncompressed O-ring

^1^ Strain rate: 238 mm/min (0.1/s) instead of 200 mm/min according to standard.

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
