# Peer review of "Analysis of O-Ring Seal Failure under Static Conditions and Determination of End-of-Lifetime Criterion"

_polymers, 2019, doi:10.3390/polym11081251_

Round 1
Reviewer 1 Report
Great and very well written paper with very interesting results.
Some remarks:
Please check your word "Figure" to be written in bold everywhere in the paper.
Since I am not very familar with the indentation test, I was asking myself why you set the force to zero after reaching the contact force of 0.03 N since the penetration force is in the regime of 0.05 N? Could you give an explanation in the paper?
On page 5, line 185: If possible, you should give a physical explanation why the intensity of the glass transition is dramatically reduced.
On page 15: The figures are too small concerning to all the others.
In the conclusions, you should point out clearly the advantages and disadvantages of your new developed method in comparison to the others you have mentioned.
I strongly recommend the paper to be published in the Journal and it was a great pleassure for me to review it!
Author Response
Dear reviewer,
thank you for your valuable advice and comments to improve the paper. Our reply is given in your review report below and respective changes are included in the revised manuscript using the “Track Changes” option.
Great and very well written paper with very interesting results.
Some remarks:
Q1: Please check your word "Figure" to be written in bold everywhere in the paper.
All„figures“are written in bold now.
Q2: Since I am not very familiar with the indentation test, I was asking myself why you set the force to zero after reaching the contact force of 0.03 N since the penetration force is in the regime of 0.05 N? Could you give an explanation in the paper?
We have clarified our explanation in the manuscript. The Instron load cell (10 N) is very sensitive and although the indenter tip is not in contact with the sample, oscillating forces (up to 0.03 N) can be observed. Therefore, the load cell is balanced and the force zeroed automatically when the tip initially gets into contact with the surface. Thus we record the force from the surface up to 0.05 N.
Q3: On page 5, line 185: If possible, you should give a physical explanation why the intensity of the glass transition is dramatically reduced.
I added to the sentence“, as the chain mobility has decreased considerably due to the increased crosslink density, and the material acts more like adensely crosslinked thermoset than a loosely crosslinked rubber”.
Q4: On page 15: The figures are too small compared to all the others.
I asked the assistant editor, this will be sorted out during article processing/layouting.
Q5: In the conclusions, you should point out clearly the advantages and disadvantages of your new developed method in comparison to the others you have mentioned.
The only method that it can be compared to is the static leakage rate measurement. As I see it, the advantages are already mentioned in the conclusions (“direct measure of seal performance”,“a more conservative criterion than the point of static seal failure“,“can be adapted to different materials, geometries and application conditions“and there are no inherent disadvantages, except perhaps the more complicated test set-up.Additionally, I added text in the introduction,in the beginning of section 3.6 and in the conclusions to highlight the advantage that our determined criterion is directly correlated to the occurrence of leakage.
I strongly recommend the paper to be published in the Journal and it was a great pleasure for me to review it!

Reviewer 2 Report
This is a very interesting piece of work. EPDM and HNBR O-rings were aged compressed at 125 °C and at 150 °C for up to 1.5 years and submitted to leak-tightness tests. The main purpose was to establish an end-of-lifetime criterion for EPDM seals. The time and amount of experimental work is very much relevant.
The manuscript is well organized and well written but it is too long. This reviewer is convinced its length could be reduced without losing relevant information, since there are excessive repetitions. Moreover, no final conclusions were drawn for the HNBR O-rings and no clear reasons are provided justifying its inclusion in this study.
It would be useful for the readers to have a detailed table summarizing all the executed tests. How many samples each seal label (Tab 1) correspond?
Line 108: “…so that a new sample was used 108 for each aging time.” What do you mean exactly?
Line 123: “…the sample thickness did not influence the measurements.” Are you assuming no size effect? These assumptions are based on what or which reference work?
Line 140: “…back in the ovens for further aging without changing the compression.” What do you mean? Is this coherent with claim (line 169) of “it was recompressed to 25 % and continued aging.”?
Lines 211-3: “However, DLO effects in the thicker O-rings prevent homogeneous and overall degradation, in contrast to the 2 mm sheet material used for the DMA measurements.” This needs to be clarified, i.e. more specific.
Line 279: “…strongly degraded with a compression set near 100 %...” What does it means 100% compression? This deserves a proper explanation.
Author Response
Dear reviewer,
thank you for your valuable advice and comments to improve the paper. Our reply is given in your review report below and respective changes are included in the revised manuscript using the “Track Changes” option.
Q1:This is a very interesting piece of work. EPDM and HNBR O-rings were aged compressed at 125 °C and at 150 °C for up to 1.5 years and submitted to leak-tightness tests. The main purpose was to establish an end-of-lifetime criterion for EPDM seals.
The paper has two main purposes:
1. Analyzing and understanding O-ring seal failure mechanisms. Here, O-rings made of two different materials (HNBR and EPDM) are analyzed, so that effects from different materials can be taken into account.
2. Presenting a method that can be used to determine a conservative end-of-lifetime criterion for static O-ring seals. The EPDM seal was used as an example to demonstrate the method. As stated in the paper the presented experiment “does not have the aim of identifying a universal value for an end-of-lifetime criterion, but it suggests a method that can be used to obtain a criterion for specific application conditions”
Q2: The time and amount of experimental work is very much relevant. The manuscript is well organized and well written but it is too long. This reviewer is convinced its length could be reduced without losing relevant information, since there are excessive repetitions.
The experimental part was restructured to make it clearer and shorter.
In the Results/Discussion section, several facts have to be repeated in a different context for discussing different aspects.
Q3: Moreover, no final conclusions were drawn for the HNBR O-rings and no clear reasons are provided justifying its inclusion in this study.
See explanation above, point 1.
Q4: It would be useful for the readers to have a detailed table summarizing all the executed tests. How many samples each seal label (Tab 1) correspond?
Each seal label corresponds to one seal.I inserted “Three O-rings per material and temperature” in the experimental description and a table at the end of the experimental section.
Q5: Line 108: “…so that a new sample was used for each aging time.” What do you mean exactly?
I reformulated the passage in the manuscript to make it clearer. It means that for leakage rate measurements, always the same samples were tested at the different aging times, but for the additional compressed samples, each aging state corresponds to a different sample.
Q6: Line 123: “…the sample thickness did not influence the measurements.” Are you assuming no size effect? These assumptions are based on what or which reference work?
The assumption is that if you penetrate only 0.02 mm into the sample, it does not matter if you have 3 or 7 mm of material underneath, as this is only 0.67 %(for 3 mm) or 0.29 %(for 7 mm) of the total thickness.I changed the text to make it clearer.
Q7: Line 140: “…back in the ovens for further aging without changing the compression.” What do you mean? Is this coherent with claim (line 169) of “it was recompressed to 25 % and continued aging.”?
The two lines refer to different experiments. The first is the static leakage rate measurement. I wanted to highlight that during all the 1.5 years of aging and measurements, the compression stayed the same and the flanges were not moved. I changed the text to make it clearer.
The second line you mentioned refers to the test with the partial decompression.
Q8: Lines 211-3: “However, DLO effects in the thicker O-rings prevent homogeneous and overall degradation, in contrast to the 2 mm sheet material used for the DMA measurements.” This needs to be clarified, i.e. more specific.
Yes, I inserted an additional explanatory sentence.
Q9: Line 279: “…strongly degraded with a compression set near 100 %...” What does it mean 100% compression? This deserves a proper explanation.
100 % compression set means that the seal cannot recover at all due to degradation and remains at the compressed height (7.5 mm in this case) after dismounting. I inserted a short explanation in the text. In the reference given in this sentence, the measurement of compression set is also explained in detail.

Round 2
Reviewer 1 Report
Very well, thanks a lot!